# MF-YOLOv10: Research on the Improved YOLOv10 Intelligent Identification Algorithm for Goods

**DOI:** 10.3390/s25102975

**Published:** 2025-05-08

**Authors:** Quanwei Wang, Xiaoyang Wang, Jiayi Hou, Xuying Liu, Hao Wen, Ziya Ji

**Affiliations:** 1School of Mechanical Engineering, Taiyuan University of Science and Technology, Taiyuan 030024, China; noveldavid@126.com (Q.W.); s202312210084@stu.tyust.edu.cn (J.H.); s202312210123@stu.tyust.edu.cn (X.L.); haowen@tyust.edu.cn (H.W.); s202312210098@stu.tyust.edu.cn (Z.J.); 2Key Laboratory of Intelligent Logistics Equipment of Shanxi Province, Taiyuan University of Science and Technology, Taiyuan 030024, China

**Keywords:** automated loading and unloading machine, MF-YOLOv10, SCSA, MPDIoU, object detection

## Abstract

To enhance the accuracy of identifying parts and goods in automated loading and unloading machines, this study proposes a lightweight detection model, MF-YOLOv10, based on intelligent recognition of goods’ shape, color, position, and environmental interference. The algorithm significantly improves the feature extraction and detection capabilities by replacing the traditional IoU loss function with the MPDIoU and introducing the SCSA attention module. These enhancements improve the detection performance of multi-scale targets, enabling the improved YOLOv10 model to achieve precise recognition of goods’ shape and quantity. Experimental results demonstrate that the MF-YOLOv10 model achieves accuracy, recall, mAP50, and F1 scores of 92.12%, 84.20%, 92.24%, and 87.98%, respectively, in complex environments. These results represent improvements of 7.11%, 11.29%, 8.51%, and 9.48% over the original YOLOv10 network. Therefore, MF-YOLOv10 exhibits superior detection accuracy and real-time performance in complex working environments, demonstrating significant engineering practicality.

## 1. Introduction

With the advancement of automated, unmanned, and intelligent cargo handling in ports, terminals, and railway freight systems [1], traditional manual loading and unloading methods can no longer meet modern demands. Addressing the inefficiencies and challenges of manual operations is crucial for improving the loading and unloading efficiency. Although loaders have evolved, issues such as human–computer interaction difficulties, spatial crossover [2], uneven palletizing, and inaccurate position recognition persist. Additionally, dust and particles generated when goods fall from heights can degrade the working environment and reduce the detection accuracy. In low-light and dusty environments, accurately identifying the location and quantity of items is critical for monitoring the loader efficiency [3]. To address these challenges, researchers are increasingly integrating intelligent recognition technology with loader control systems, leveraging computer vision and machine learning to achieve automated operations and precise control. This integration significantly enhances the processing speed while reducing the labor costs.

At present, traditional manual and machine methods for detecting the position and quantity of piece goods can no longer meet the production requirements of enterprises. With the continuous development of deep learning, detection algorithms based on deep learning are becoming mainstream. Deep learning-based object detection algorithms can be divided into one-stage and two-stage algorithms based on their framework structure. One-stage object detection algorithms directly output the target bounding box and category without the need for a candidate region generation step, avoiding the complex process of candidate box generation and refinement required in two-stage methods. These mainly include YOLO [4], SSD [5], RetinaNet [6], and CenterNet [7]. One-stage methods are generally faster and more computationally efficient, making them suitable for real-time detection tasks. Two-stage object detection algorithms, on the other hand, first generate candidate regions and then perform fine classification and bounding box regression within these regions. Two-stage algorithms typically include R-CNN [8], Fast R-CNN [9], Faster R-CNN [10], and Mask R-CNN [11]. By introducing candidate region generation and fine classification, two-stage algorithms achieve higher detection accuracy, especially excelling in complex scenes and small object detection. However, due to their higher computational cost and slower inference speed, they are generally more suitable for applications where high accuracy is required but real-time performance is less critical. Scholars both domestically and internationally have conducted in-depth research on piece goods recognition technology. Ao Wang et al. [12] proposed YOLOv10, which addresses the need for post-processing through a consistent dual assignment strategy, reducing the model detection time and achieving the fastest detection speed to date. Sunwoo Hwang et al. [13] proposed an object recognition algorithm for automated loading devices to improve the performance of amorphous cargo identification. This research enhances the capability of automated loading devices in recognizing amorphous cargo. Arjun et al. [14] addressed the shape recognition issues by preprocessing images to extract the contours of assembled piece goods and normalizing the shape descriptions using the equal-area method. Although this method can improve the detection accuracy under certain conditions, it still faces significant challenges in complex and variable real-world application scenarios. Ke jia Li et al. [15] designed a simple and easy-to-operate railway boxcar piece goods loading and unloading machine, analyzing the stability issues during loading and unloading operations under different postures, but did not mention the impact and detection of falling goods during operations. Jeon-Seong Kang et al. [16] proposed an image enhancement method for recognizing the package loading status under various lighting conditions, which can address insufficient lighting inside trucks. Although this method effectively improves the image quality, further research is needed for optimization in complex scenes and integration with other systems. Lijun Gou et al. [17] proposed a local surface segmentation algorithm that improves the detection accuracy by extracting the skeleton of stacked cartons and reconstructing invisible contours. This method effectively reduces the need to collect a large number of foreground instances but still requires further improvement in terms of adaptive brightness adjustment and reducing pixel artifacts. Sitong Guan et al. [18] proposed a method for accurately detecting and counting wheat spikes by combining the BiFPN, SEAM, and GCNet modules. This method enhances the feature extraction and detection capabilities, enabling precise target recognition under various environmental conditions and providing strong support for the precise control of automatic loading and unloading machines. Chicco, Davide, et al. [19] contributed to the critical factors when selecting appropriate performance metrics for regression analysis by comparing the widely used coefficient of determination and symmetric mean absolute percentage error, demonstrating that the coefficient of determination (R) is more informative and authentic than the SMAPE. T.W. Rogers et al. [20] analyzed each small window of the image separately and detected loads within the window by random forest classification of the texture features together with the window coordinates.

While existing approaches have improved the defect detection performance in parcel images, significant challenges remain for practical applications. Current detection algorithms often fail to adequately balance accuracy with computational efficiency—many models exhibit insufficient parameter reduction or excessive computational demands, compromising their viability for resource-constrained edge devices. Furthermore, parcels present unique detection complexities due to their dense arrangements, irregular placement patterns, and variable dimensions [21]. These challenges are exacerbated in high-density stacked parcel scenarios where the boundary ambiguity and interference from rectangular background objects frequently lead to false positives and missed detections. Building upon these disadvantages, we present MF-YOLOv10—an enhanced YOLOv10-based algorithm for robust parcel detection. Our principal contributions include the following:(1)We built a well-annotated parcel dataset that simulates real-world scenarios and conditions. The dataset includes parcels captured from various angles, under different lighting conditions and resolutions, with a particular focus on railway freight yard scenarios. Image enhancement techniques were applied to further improve the dataset’s generalization capability, comprehensively covering different parcel arrangement patterns under various working conditions to enable accurate parcel counting across multiple scenarios.(2)MPDIoU loss function: Specifically designed to enhance the detection of overlapping, blurred, and small objects—particularly those near image edges—by improving the model convergence speed and localization accuracy.(3)SCSA attention mechanism: Effectively focuses on target regions while suppressing complex backgrounds, significantly boosting the small object detection capability.(4)We conducted extensive experiments on the parcel dataset detection task. The results demonstrate that compared with other detection algorithms, our proposed MF-YOLOv10 model not only improves the detection accuracy but also significantly reduces the model size and enhances the inference speed, providing robust technical support for real-time parcel counting and detection on embedded platforms.

## 2. Automatic Loading and Unloading Machine Identification Scheme

### 2.1. Basic Principles of Loading and Unloading Machines

The loader is designed for loading and unloading goods in railway boxcars. The maximum weight of a single cargo piece is 50 kg, and the maximum width is 800 mm. During loading and unloading, the loader’s telescopic belt conveyor must be able to easily enter the car and rotate freely within it, without the need to raise the outriggers or move the loader’s position. The orientation of the input and output of the loading and unloading machine can be adjusted according to the goods’ position. Additionally, the loader should be capable of moving flexibly on the platform while maintaining stability during loading and unloading operations. To meet these requirements, a multifunctional, integrated automatic loading and unloading machine has been designed. This system includes a telescopic rotary mechanism, a lower rotary table, an upper rotary table, a discharging belt conveyor, a feeding belt conveyor, a hydraulic system, and an electronic control system. A 3D model of this multifunctional all-in-one automatic cargo loading and unloading machine is shown in Figure 1.

### 2.2. YOLOv10 Algorithm

As a first-stage algorithm, YOLOv10 pays more attention to small object detection than other YOLO series object detection algorithms, and it also pays more attention to the network architecture, inference, deployment capability, detection accuracy, and speed. According to the model size, YOLOv10 can be divided into five different widths and depths: s, m, l, x and n, which can meet the requirements of various datasets. Considering the limited storage resources of edge devices in practical parcel detection scenarios, this study adopts YOLOv10n as the baseline model for optimization. Our approach aims to enhance the detection accuracy for parcel objects while simultaneously reducing the model parameters.

From the perspective of network structure optimization, the detection model is generally divided into three parts: backbone, neck, and head. The backbone is responsible for feature extraction, the neck is used for feature fusion, and the head is used for object classification and positioning. The lightweight design is a key metric for edge device detection models, designed to reduce the model size while maintaining good accuracy with limited computing resources. The model structure is shown in Figure 2.

## 3. MF-YOLOv10 Network

In recent years, object detection and image recognition technologies have been widely adopted in the industrial sector, particularly with the rise of deep learning. Object detection algorithms based on convolutional neural networks have advanced rapidly. Among these, the YOLO series models have become the mainstream choice for industrial applications due to their end-to-end training methods and efficient inference capabilities. Earlier versions of YOLO were used for target recognition in industrial equipment such as forklifts and cranes [22]. As a next-generation object detection algorithm, YOLOv10 builds on the high efficiency of previous YOLO models while optimizing both the model accuracy and the inference speed. This makes it especially suitable for real-time object detection in industrial scenarios. YOLOv10 introduces new architectural optimizations and loss function designs, resulting in significant improvements in the detection accuracy and speed. These advancements are particularly beneficial for applications requiring real-time detection and control, further advancing the field of intelligent goods identification.

### 3.1. Research Work

We performed the following work: (1) dataset construction, (2) annotation, (3) augmentation, and (4) model training.

*(1)* 
*Dataset Construction*


For the railway freight scenario, 3000 images of goods were captured using CCD cameras. The image collection process ensured that the dataset accurately reflects the characteristics of a real loading and unloading environment, including the irregular distribution, uneven lighting, and occlusion between goods. Data collection was conducted between 8:00 a.m. and 2:00 p.m., ensuring a diverse and widely distributed dataset, which enhances the generalization ability of the improved model and aligns with the requirements of real-world scenarios. To train the YOLOv10 model, a dedicated dataset was created specifically for loading and unloading operations. This dataset is shown in Figure 3.

*(2)* 
*Dataset Annotation*


This study was annotated using Labellmg software (version 1.8.6) in txt format.

The dataset includes image samples of goods with various shapes and sizes encountered during loader operations, such as flour bags, rice bags, and cartons. Each image was manually labeled with a bounding box and category label corresponding to the object identified by the loader. This is illustrated in Figure 4.

*(3)* 
*Dataset Augmentation*


Prior to the model training, we implemented an extensive data augmentation pipeline to address three critical objectives: maximizing the dataset utilization without additional labeling effort, preventing model overfitting and enhancing the generalization capabilities. Through systematic application of augmentation techniques to each annotated image, we expanded our original dataset from 3000 to 9000 training samples—a threefold increase that significantly improved the feature diversity.

Our augmentation strategy incorporated three complementary approaches: geometric transformations: random rotation (±10° range); multi-scale resizing (0.5~1.5 scaling factors); photometric adjustments: hue variation (±0.015 delta); dynamic saturation adjustment (0.7~1.3 scaling); controlled Gaussian noise injection (σ = 0.01); spatial modifications: adaptive random cropping (3%~15% image area removal).

*(4)* 
*Model Training*


Based on the pre-trained YOLOv10n model, the 9000 images were then divided into a training dataset, a test dataset, and a validation dataset in an 8:1:1 ratio. The model was trained with a batch size of 64, optimized using SGD with momentum (0.8) and weight decay (0.0001) to enhance the training stability and generalization performance.

### 3.2. Piece Inspection Network Structure

This paper proposes an improved MF-YOLOv10 network structure based on the YOLOv10 model. The improvements include replacing the traditional IoU loss function with the MPDIoU and introducing an attention module, the SCSA, which enhances the detection performance of targets at various scales. Sparse convolutions were employed to handle partially occluded inputs, improving the feature diversity while boosting the computational efficiency and reducing the memory usage. The SCSA attention mechanism was integrated on top of the SPFF layer to enhance the learning of the attention relationships between network channels, thereby improving the detection accuracy of adjacent and occluded objects. Finally, the IoU was optimized using angle and distance compensation in the SIoU loss function, enabling the model’s predicted bounding box to be quickly and accurately located. The improved network structure is shown in Figure 5.

### 3.3. MPDIoU Loss Function

The MPDIoU is a loss function designed for bounding box regression tasks in object detection [23], aimed at enhancing the model’s ability to learn the position of the target box. It improves the model’s robustness when handling objects that have less overlap or are more distant by introducing distance information between bounding boxes.

The traditional IoU loss calculates the overlap between the target box and the ground truth box but performs poorly in detecting small objects or cases with subtle overlap. The principle is illustrated in Figure 6. The mathematical expression is given by Equation (1). Many existing bounding box regression loss functions yield the same value for different predictions, which slows down the convergence and reduces the accuracy of the bounding box regression. To address this, the MPDIoU is introduced as a new metric to compare the similarity between the predicted bounding box and the ground truth box during the regression process, taking into account the strengths and weaknesses of existing BBR loss functions.(1)$$IoU=\frac{A\cap B}{A\cup B}$$

Building on the traditional *IoU*, the *MPDIoU* introduces three key parameters: the distance between the centers of the predicted and real boxes, the aspect ratio of the bounding box, and the overlap of the regions. The center point distance reflects the relative position between the centers of the predicted and actual boxes; the bounding box aspect ratio helps the model learn a more accurate box shape; and the region overlap uses the IoU to calculate the ratio of overlapping areas between the two boxes, combining these factors into a comprehensive loss function.

Formula for the *MPDIoU* loss function: The specific formula is represented as Equations (2)–(4).(2)$$d_{1}^{2}=(x_{1}^{p}-x_{1}^{g}{)}^{2}+(y_{1}^{p}-y_{1}^{g}{)}^{2}$$(3)$$d_{2}^{2}=(x_{2}^{p}-x_{2}^{g}{)}^{2}+(y_{2}^{p}-y_{2}^{g}{)}^{2}$$(4)$$MPDIoU=IoU-\frac{d_{1}^{2}}{h^{2}+w^{2}}-\frac{d_{2}^{2}}{h^{2}+w^{2}}$$

In Figure 7, *h* and *w* represent the length and width of the image to be detected, the yellow rectangle represents the real frame, the red rectangle represents the prediction frame, ($x_{1}^{g},y_{1}^{g}$) and ($x_{2}^{g},y_{2}^{g}$) represent the coordinates of the upper left and lower right corners of the real box, and ($x_{1}^{p}$, $y_{1}^{p}$) and ($x_{2}^{p}$, $y_{2}^{p}$) represent the coordinates of the upper left and lower right corners of the prediction frame, respectively.

Thanks to the introduction of the center point distance and shape factors, the *MPDIoU* performs better in complex scenes and small object detection. Compared with the traditional *IoU* loss, the *MPDIoU* can effectively reduce the loss oscillation during the training process and make the model converge faster. The *MPDIoU* loss function can significantly improve the effect of the object detection task by considering the overlap of the bounding box, the distance between the center points and the shape of the box. It is a powerful complement to the traditional *IoU* losses and especially excels when dealing with complex targets.

### 3.4. SCSA Attention Mechanism

In this study, the SCSA attention mechanism [24] was strategically integrated atop the SPPF layer of the backbone network, a design choice driven by two key considerations. (1) Architectural synergy—the SPPF layer’s multi-scale feature outputs provide an optimal stage for the SCSA’s dual-path attention to filter noise and enhance small object detection, as evidenced by our ablation studies showing a 2.3% mAP50 improvement over alternative placements. (2) Computational efficiency—processing features post-SPPF saves 0.7 ms inference time by avoiding redundant attention calculations. This enhanced architecture significantly improves the global feature representation learning while effectively suppressing the interference from complex backgrounds. The SCSA attention mechanism preserves the original feature dimensionality, maintaining richer object feature information and thereby strengthening the representation capability. Through local interactions enabled by 1D convolution, the model dynamically focuses on key regions relevant to the loading/unloading accuracy and automatically adjusts the receptive field ranges across different feature layers, providing flexible adaptation to variations during palletizing operations.

The SCSA attention mechanism module is designed to improve the feature representation capability of deep learning models, particularly in tasks such as object detection and image segmentation. The SCSA combines spatial attention and channel attention, aiming to enhance important features and suppress irrelevant ones, thereby boosting the model’s performance. The SCSA module primarily consists of three components: channel squeeze-and-excitation, spatial attention, and feature recalibration. The operational pipeline for input features with the dimensions (B = 1, C = 32, H = 256, W = 256) is as follows. Squeeze: Performs global average pooling (GAP) on the input feature maps to generate channel descriptors. This process computes the global information for each of the 32 channels (dim = 32) to capture the inter-channel relationships. Excitation: Utilizes two fully connected layers (with a reduction ratio r = 16 by default) and activation functions (ReLU → Sigmoid) to produce the channel-wise weights, dynamically adjusting the importance of each channel. For the given dim = 32 configuration, this creates a compact embedding of size 32/r = 2 before re-expanding. Spatial attention: A 7 × 7 convolutional kernel (window_size = 7) to generate spatial attention maps; multi-head attention mechanism (head_num = 8) for parallel spatial relationship modeling; sigmoid activation to emphasize important regions in the 256 × 256 feature space. Feature recalibration: Combines the outputs through element-wise multiplication: Channel weights (1 × 1 × 32) × spatial weights (256 × 256 × 1); enhances the discriminative features while suppressing irrelevant activations; maintains the original dimensions (1,32,256,256) for seamless integration. Channel compression and excitation: Compression: Global average pooling is applied to the input feature map to generate channel descriptors. This process captures the relationships between the channels by computing the global information for each channel. Excitation: The channel weights are generated through fully connected layers and activation functions, dynamically adjusting the importance of each channel. Spatial attention: The feature map after channel excitation is processed using convolutional operations to generate a spatial attention map. This process emphasizes important spatial locations in the feature map by focusing on different regions. Feature recalibration: The results of the channel and spatial attention are combined, and the original feature map is recalibrated to produce the final feature output. This is achieved through element-wise multiplication, amplifying important features while suppressing irrelevant ones.

By simultaneously focusing on channel and spatial information, the SCSA can more effectively extract important features, thereby improving the model’s recognition accuracy. Additionally, it can adaptively adjust the weights of different channels and spatial positions in the feature map, making it suitable for various tasks and datasets. Compared to other complex attention mechanisms, the SCSA is computationally more efficient, making it suitable for lightweight models. By incorporating the SCSA, the model can achieve better performance under various conditions, enhancing the overall effectiveness of intelligent recognition systems. Figure 8 shows the schematic of the SCSA attention mechanism.

### 3.5. Model Evaluation Index

In this experiment, five indicators, including the precision, recall, mean average precision (mAP50), mean average *precision* at IoU thresholds from 0.5 to 0.95 with 0.05 increments (mAP50:95) and F1 score, were used to measure the detection effect of the improved YOLOv10 model. The formula is shown below in (5)–(9):(5)$$Precision=\frac{TP}{TP+FP}\times 100\%$$(6)$$Recall=\frac{TP}{TP+FN}\times 100\%$$(7)$$AP=\int_{0}^{1}P(R)dR$$(8)$$mAP=\frac{\sum_{j=1}^{c}(AP{)}_{j}}{c}$$(9)$$F1-score=2\times \frac{Precision\times Recall}{Precision+Recall}$$
where *TP* represents the number of positive samples predicted correctly, *FP* represents the number of other samples predicted as positive, and *FN* represents the number of positive samples predicted as other samples. The precision reflects the model’s ability to distinguish between negative samples, and the higher the precision, the more accurate the model is in predicting positive samples. The higher the *recall*, the higher the proportion of positive samples correctly identified by the model. The *F1 score* is a combination of the two, and the higher the value, the more stable the model. *C* in Equation (7) indicates the number of classes tested (4 in this experiment). When calculating the *mAP50*, the IoU of the model is set to 0.5, that is, when the intersection union ratio between the real frame and the prediction frame is greater than 0.5, the prediction of the model is considered correct. When there is a large gap between the accuracy and the recall, the F1 score can provide a more comprehensive evaluation of the model’s performance.

At the same time, we measured the counting performance of the model using the $R^{2}$, mean absolute percentage error (*MAPE*), mean absolute error (*MAE*), and *RMSE* as evaluation metrics. The $R^{2}$ measures the degree of regression, indicating how closely the predicted piece count correlates with the actual count. A high $R^{2}$ value indicates a better model fit, reflecting the proportion of variation in the actual piece count explained by the predicted count. The *MAPE* calculates the average of the absolute percentage error, measuring the percentage error between the forecast and the actual piece count. The MAE represents the average of the absolute error between the predicted and actual piece counts, representing the mean deviation. The *RMSE* is the square root of the mean square error and provides a measure of the margin of error. Lower values for the *MAPE*, *MAE*, and *RMS* indicate higher prediction accuracy, and by combining these metrics, we can comprehensively evaluate the model’s counting performance from different perspectives. These evaluation metrics are defined by Equations (10)–(13), where $t_{i}$ is the true value of the i-th sample, $p_{i}$ is the predicted value of the i-th sample, $\overset{-}{t_{i}}$ is the average of the true values of all samples, and *n* is the total number of samples.(10)$$MAPE=\frac{1}{n}\sum_{i=1}^{n}|\frac{t_{i}-p_{i}}{t_{i}}|\times 100\%$$(11)$$MAE=\frac{1}{n}\sum_{i=1}^{n}|t_{i}-p_{i}|$$(12)$$RMSE=\sqrt{\frac{\sum_{i=1}^{n}(t_{i}-p_{i}{)}^{2}}{n}}$$(13)$$R^{2}=1-\frac{\sum_{i=1}^{n}(t_{i}-p_{i}{)}^{2}}{\sum_{i=1}^{n}(t_{i}-\overset{-}{t_{i}}{)}^{2}}$$

## 4. Experimental Results and Analysis

### 4.1. Experimental Environment Configuration and Network Parameters

In order to validate the effectiveness of the method in this article, a relevant experimental platform was built. The experimental computer was configured with a 64-bit Windows 11 operating system, under the deep learning framework of PyTorch version 2.0.1, using the Python programming language version 3.10.6, with an image size of 640 × 640, and an initial learning rate of 0.01 to balance the model’s convergence speed and learning efficiency, preventing instability due to rapid convergence. The experimental parameter configuration is shown in Table 1 below.

The visualization results of the dataset from Figure 9 show that the distribution of the locations of various items is uniform. From the relative width and height of the labels, it can be seen that most labels have a relative width between 0 and 1.0, and a relative height between 0 and 0.4. The sizes are relatively small, falling into the category of small targets, which is also one of the difficulties affecting item detection performance.

### 4.2. Improve Algorithm Experiments

In order to evaluate the performance of the MF-YOLOv10 intelligent identification system for loaders, we designed a series of experiments to verify its detection accuracy, real-time performance and robustness in complex working environments. The experimental environment included different scenarios, such as open factories and storage areas, and the detection targets were mainly goods.

In this paper, for the box loss, cls loss, and dfl loss curves of the training set, the validation set, and the other performance indicators of the precision, recall, mAP50, and mAP50:95, the convergence curves are shown in Figure 10, and the abscissa represents the number of iterations, and the experiment has a total of 50 iterations. As can be seen from Figure 9, after 45 iterations of the model, each loss curve tended to be close to 0, and the model entered convergence. Eventually, the box loss value was around 0.

The improved algorithm further integrates the location and semantic information of the pieces in the enclosed space, and it can extract the three-dimensional features of the pieces, so as to accurately detect the location and quantity of the pieces in complex environments.

### 4.3. Model Count

We further analyzed the algorithm counting function of MF-YOLOv10 and randomly selected 50 model test images in a large number of different complex environments. The algorithm had the ability to calculate the piece-to-shipment peaks in a variety of complex environments by randomly selecting 50 images for model testing. The results are shown in Figure 11 and Table 2. The $R^{2}$ values for the number of pieces detected using YOLOv6, YOLOv7, YOLOv8, YOLOv9, and YOLOv10 were 0.58, 0.63, 0.68, 0.71, and 0.78, respectively, compared to the manual counting results. The MPE values were 17.36%, 16.95%, 14.82%, 12.68%, and 11.21%, the MAPE values were 14.30%, 12.85%, 10.67%, 8.56%, and 5.69%, and the RMSE values were 18.56%, 16.81%, 15.43%, 14.38%, and 10.83%, respectively. In contrast, the MF-YOLOv10 algorithm achieved an $R^{2}$ of 0.86, an MAPE of 4.63%, an MAE of 9.62%, and an RMSE of 8.61% compared to the manual counting results. An increase in the $R^{2}$ value indicates a higher correlation with the improved algorithm.

### 4.4. Ablation Experiment of MF-YOLOv10 Algorithm

In order to further verify the impact of the improved MF-YOLOv10 on the performance of the item inspection model, and to verify the effectiveness of each improvement proposed in this paper, we tested four scenarios: no module, no SCSA module, no MPDIoU module, SCSA and MPDIoU modules, where each model had the parameters based on the YOLOv10 model and the training configuration was the same as before. The results of the experiment are shown in Table 3. The modified algorithm was systematically compared with the initial algorithm to evaluate the impact of each improvement. “+” means that the original structure had not changed, and “−” means that modules were added to the original structure.

As shown in Table 3, the precision, recall, mAP50, and mAP50:95 of YOLOv10 were 74.01%, 60.91%, 71.73%, and 44.23%, respectively, without any additional modules. After introducing the SCSA module, the precision, recall, mAP50, and mAP50:95 of the YOLOv10 model improved by 2.42%, 0.44%, 1.95%, and 1.72%, respectively, compared to the baseline YOLOv10 network. Following the addition of the MPDIoU module, the precision, recall, mAP50, and mAP50:95 increased by 3.34%, 0.16%, 2.25%, and 1.81%, respectively, over the baseline YOLOv10 network. When both the SCSA and MPDIoU modules were combined, the precision, recall, mAP50, and mAP50:95 improved by 10.27%, 5.83%, 7.18%, and 7.39%, respectively, compared to the baseline YOLOv10 network.

### 4.5. Comparative Experiments

After integrating the SCSA and MPDIOU modules, we conducted comparative experiments with the improved YOLOv10 model with the YOLOv5, YOLOv6, YOLOv7, YOLOv8, YOLOv9 and original YOLOv10 models. A total of 900 images were used as validation sets.

From the results in Table 4, it can be seen that the proposed model has obvious advantages in terms of the precision, mAP50, params and GFLOPs. In terms of the detection precision, the proposed model demonstrates superior performance, with improvements of 10.11%, 9.29%, 8.5%, 10.75%, and 7.11% over YOLOv5, YOLOv6, YOLOv7, YOLOv8, YOLOv9, and YOLOv10, respectively, indicating that MF-YOLOv10n performs better in detection accuracy. In terms of the model complexity, the parameter params of MF-YOLOv10n is lower than that of the other seven models, and the weights occupied by the weight file are the lowest among all the models. The low complexity and weight file size have good accuracy at the same time, which is suitable for the inspection task of railway freight yards. In terms of the average detection speed, the MF-YOLOv10n model has an FPS of 84.7, which is 62.7, 69, 75.6, 56.9, and 39.9 higher than YOLOv5, YOLOv6, YOLOv7, YOLOv8, and YOLOv9, and lower than 26.3 compared to YOLOv10n. Although YOLOv10 achieves a higher FPS (111 vs. 84.7) compared to MF-YOLOv10, it underperforms in all the other key metrics. In parcel detection scenarios, higher precision and mAP50 take precedence over FPS, making MF-YOLOv10′s superior accuracy–speed balance more practical and valuable. Although YOLOv8 has high accuracy, the FPS is low, which cannot meet the real-time requirements. In order to visually demonstrate the detection effect of the MF-YOLOv10n algorithm, MF-YOLOv10n and five other models with better average accuracy were selected for visual comparison of the detection results, as shown in the figure. The results show that the improved MF-YOLOv10 model performs excellently at 92.12%, 84.20%, 92.24% and 64.90% in terms of the accuracy, recall, average detection accuracy and mAP50:95, respectively, indicating that MF-YOLOv10 has high accuracy in detecting goods.

### 4.6. Detect the Effect

In order to verify the detection performance of the model, the YOLOv10 model and the MF-YOLOv10 model were used to test the pieces of goods images in the test set. Figure 12 shows the detection effect of the MF-YOLOv10 model for various defects. It can be seen that the MF-YOLOv10 model has a significant improvement in the detection confidence of the location of the piece.

In addition, MF-YOLOv10 stands out for its excellent balance between the number of parameters and the computational complexity, reducing the model parameters by 7.43% while improving the detection accuracy by 7.11% compared to the original YOLOv10 algorithm. This enhancement is important for industrial applications. By integrating various improvements, several experiments have confirmed that our method is particularly suitable for the inspection of goods in railway boxcars, and the results have demonstrated the clear advantages of our model in terms of all the performance indicators. Together, when all the improvements are combined, their effects are further amplified, significantly enhancing the overall performance of piece location detection without sacrificing real-time processing power.

## 5. Conclusions

In this study, we improved the YOLOv10 algorithm and proposed the MF-YOLOv10 detection algorithm to improve its accuracy and efficiency in the task of piece identification and detection. The core contents of the improvement include the introduction of the new feature extraction module SCSA, the optimization loss function MPDIoU, and the improvement of the model’s multi-scale detection ability. By comparing the experimental results, it is shown that these improvements can effectively improve the performance of YOLOv10 on different datasets.

The experimental results show that the improved YOLOv10 algorithm has significantly improved the accuracy and recall rate, compared with the original YOLOv10 model, the mAP50 is increased by about 8–9 percentage points, and the inference speed is also optimized. In conclusion, our improved scheme effectively improves the performance of YOLOv10 in handling complex scenes and small target detection tasks, which proves the feasibility and application value of the proposed method. Future research will continue to optimize the deployment of the model on edge devices and further improve the adaptability to multi-target and multi-condition environments.

## Figures and Tables

**Figure 1 sensors-25-02975-f001:**
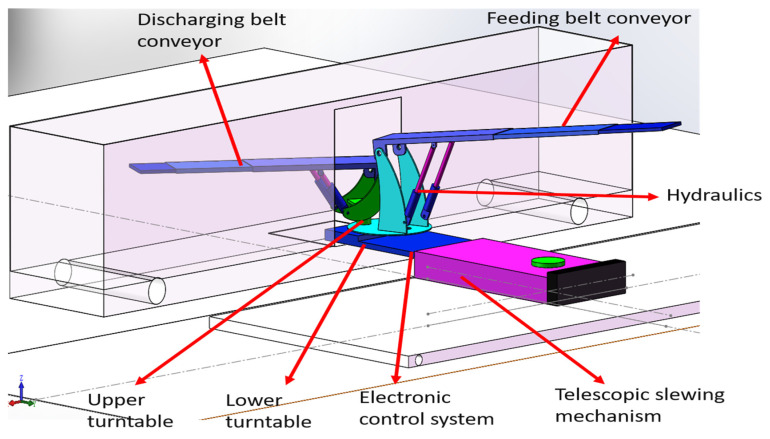
Piece loading and unloading machine 3D model.

**Figure 2 sensors-25-02975-f002:**
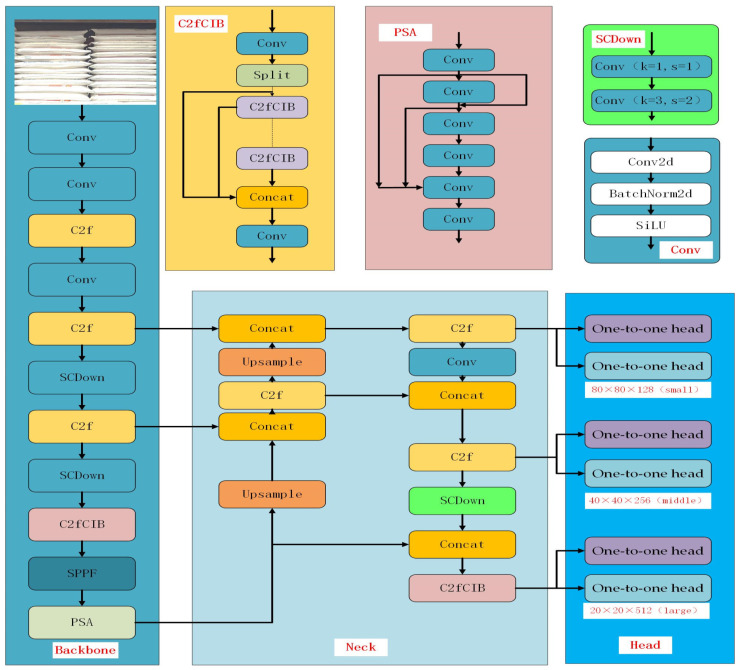
YOLOv10 network diagram.

**Figure 3 sensors-25-02975-f003:**
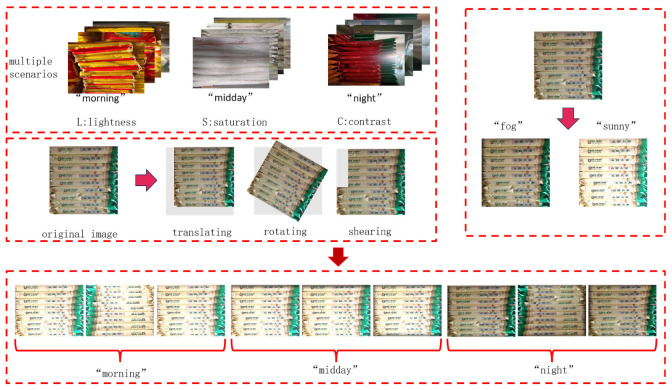
Goods dataset.

**Figure 4 sensors-25-02975-f004:**
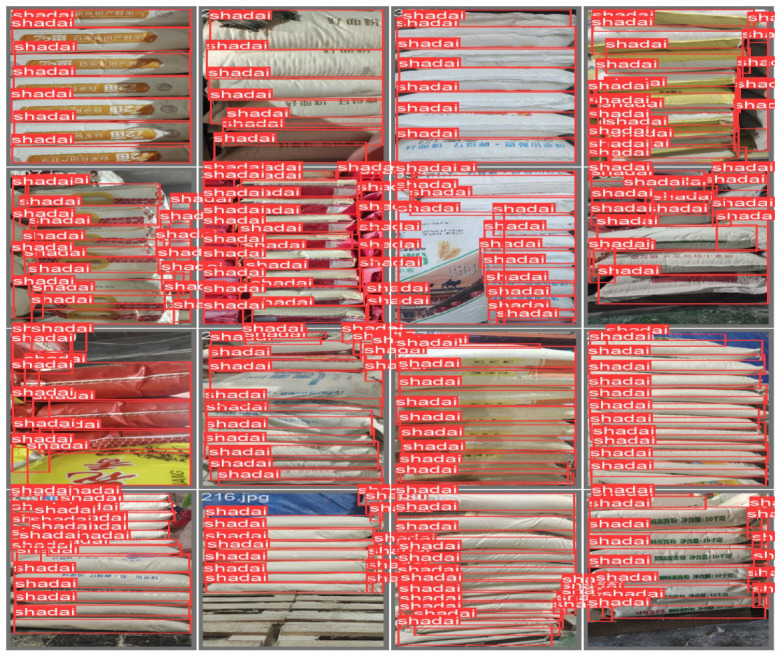
Dataset labeling.

**Figure 5 sensors-25-02975-f005:**
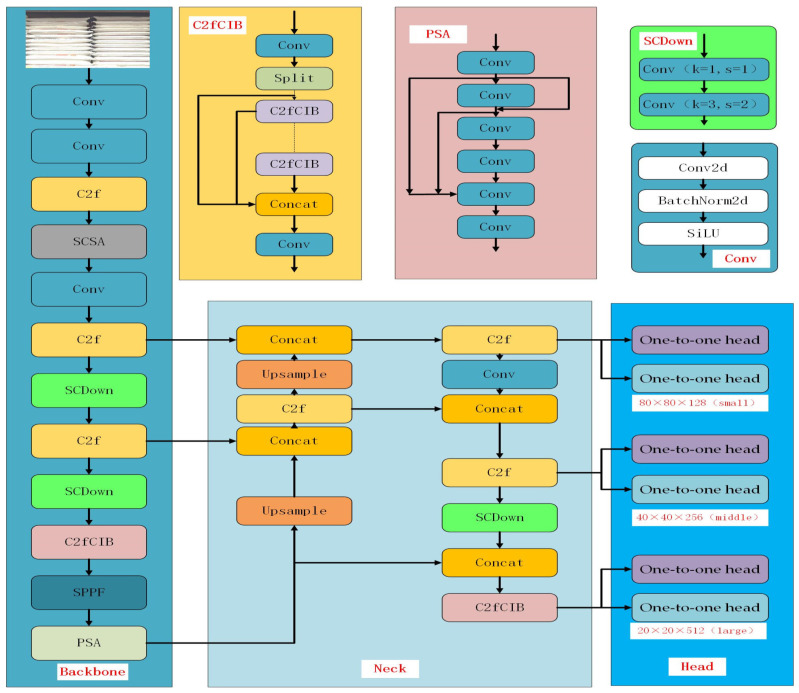
The improved YOLOv10 network architecture.

**Figure 6 sensors-25-02975-f006:**
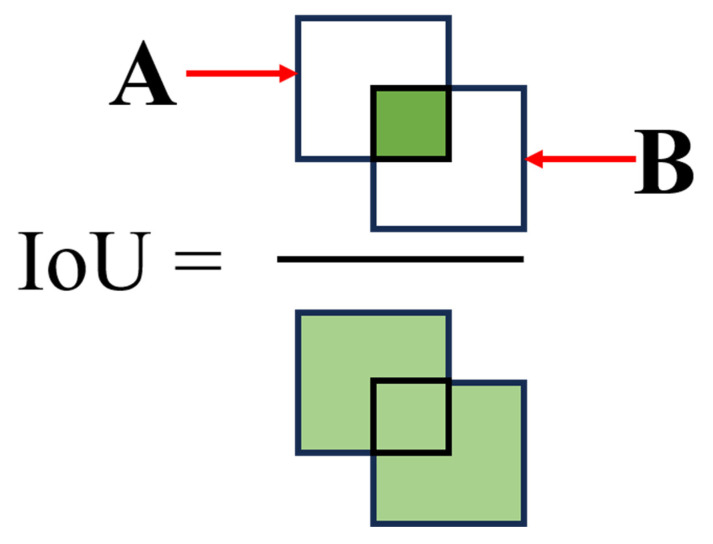
Definition of the IoU.

**Figure 7 sensors-25-02975-f007:**
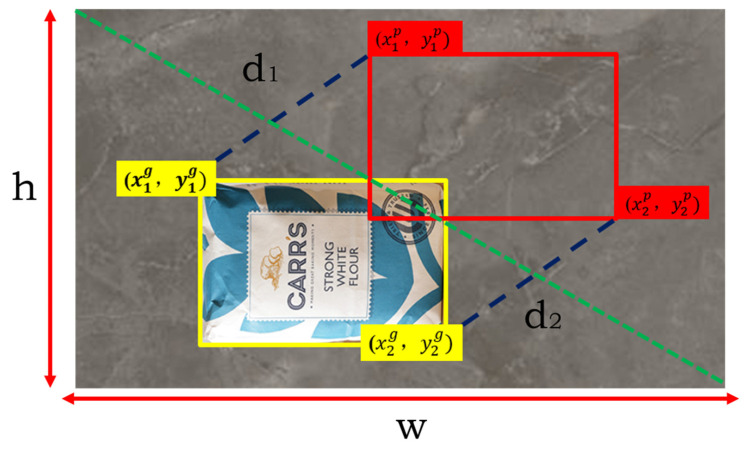
Schematic diagram of the MPDIoU function.

**Figure 8 sensors-25-02975-f008:**
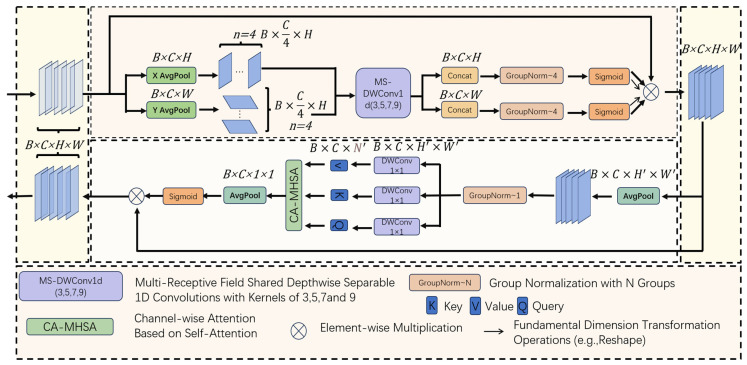
SCSA attention mechanism.

**Figure 9 sensors-25-02975-f009:**
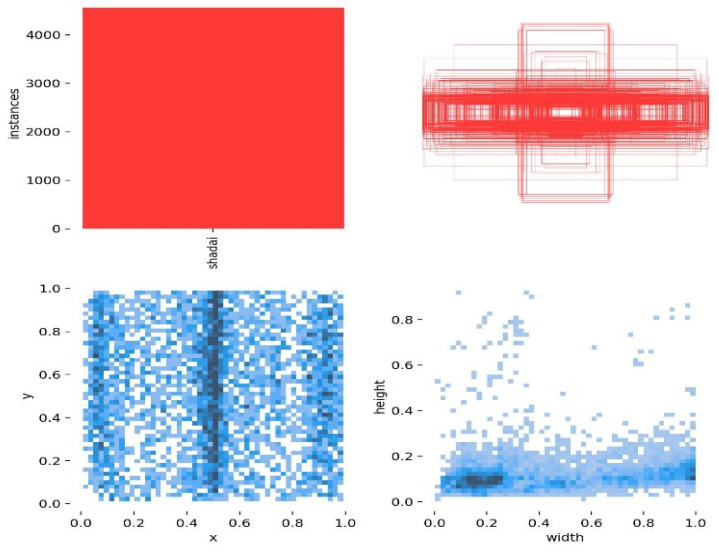
Label location visualization in the model.

**Figure 10 sensors-25-02975-f010:**
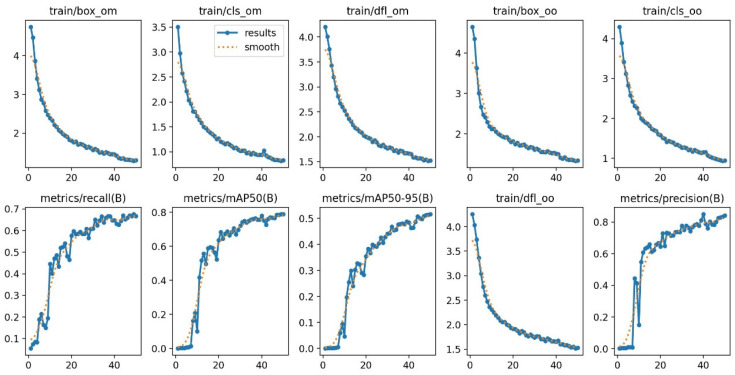
Loss function convergence curve of MF-YOLOv10.

**Figure 11 sensors-25-02975-f011:**
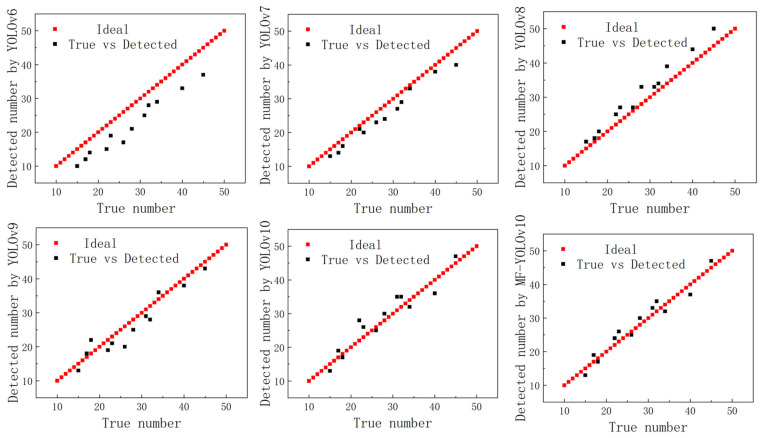
Comparison of the counting results.

**Figure 12 sensors-25-02975-f012:**
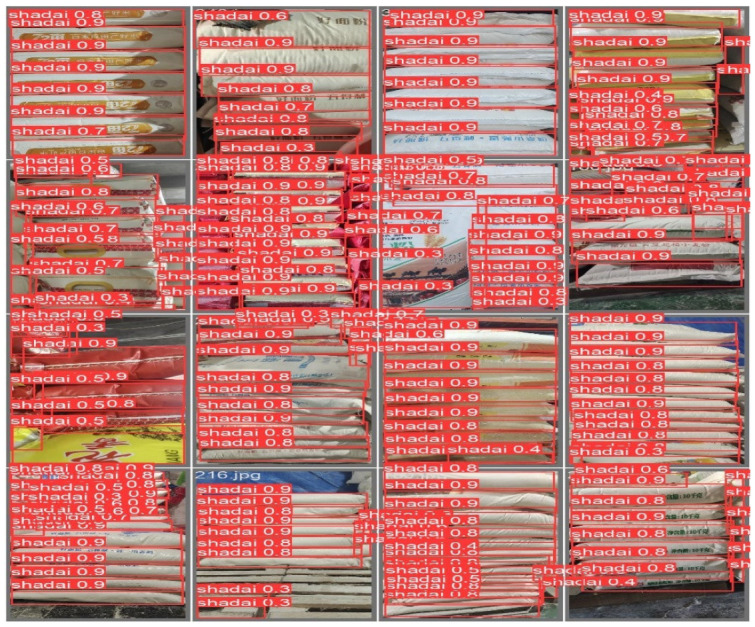
Object detection visualization.

**Table 1 sensors-25-02975-t001:** Experiment parameter configuration.

Disposition	Performance Parameters
Operating system	Windows 11 64-bit
CPU	Intel(R) Xeon(R) Platinum 8457C
GPU	A100
RAM	32 GB
Image size	640 × 640
The number of iterations	50
Learning rate	0.01

**Table 2 sensors-25-02975-t002:** Count values for the different versions.

Algorithm	R	MAE	MAPE	RMSE
YOLOv6	0.58	17.36%	14.30%	18.56%
YOLOv7	0.63	16.95%	12.85%	16.81%
YOLOv8	0.68	14.82%	10.67%	15.43%
YOLOv9	0.71	12.68%	8.56%	14.38%
YOLOv10	0.78	11.21%	5.69%	10.83%
MF-YOLOv10	0.86	9.62%	8.61%	4.63%

**Table 3 sensors-25-02975-t003:** Ablation test results.

Algorithm	SCSA	MPDIoU	Precision	Recall	mAP50	mAP50:95
YOLOv10	−	−	74.01	60.91	71.73	44.23
YOLOv10	+	−	76.43	61.35	73.68	45.95
YOLOv10	−	+	77.35	61.07	73.98	46.04
MF-YOLOv10	+	+	84.28	66.74	78.91	51.62

**Table 4 sensors-25-02975-t004:** Experiments comparing the different versions.

Algorithm	Precision	Recall	mAP50	mAP50:95	Params	GFLOPs	FPS
MF-YOLOv10	92.12	84.20	92.24	64.90	2,506,394	7.2	84.7
YOLOv10	85.01	72.91	83.73	56.23	2,707,430	8.4	111
YOLOv9	81.37	64.53	75.85	44.03	9,743,366	39.6	44.8
YOLOv8	92.89	84.77	93.11	64.90	3,011,043	8.2	27.8
YOLOv7	83.62	62.36	73.69	45.32	4,185,693	10.5	9.1
YOLOv6	82.83	70.74	81.26	49.63	4,238,243	11.9	15.7
YOLOv5	82.01	65.91	77.73	50.23	2,508,659	7.2	22

## Data Availability

The data used in this study can be obtained from the second author at wxy3130126@163.com following a reasonable request.

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
