# Peer review of "MF-YOLOv10: Research on the Improved YOLOv10 Intelligent Identification Algorithm for Goods"

_sensors, 2025, doi:10.3390/s25102975_

Round 1
Reviewer 1 Report
Comments and Suggestions for Authors
This study presents an enhanced YOLOv10-based model, MF-YOLOv10, tailored for intelligent goods detection in automated loading/unloading systems, incorporating innovations such as the MPDIoU loss function and the SCSA attention mechanism. The work addresses a relevant industrial challenge and demonstrates improvements in detection accuracy and real-time performance. Below, I provide a detailed evaluation of the manuscript’s strengths, weaknesses, and suggestions for improvement.
1: It is noted that your manuscript needs careful editing by someone with expertise in technical English editing paying particular attention to English grammar, spelling, and sentence structure so that the goals and results of the study are clear to the reader.
2: The equation numbering in the manuscript is inconsistent in alignment. Currently, equation labels (e.g., "(1)", "(2)") are not right-aligned, which deviates from standard academic formatting conventions. For example: Equation (1)-(3) under Section 3.3 (MPDIoU Loss Function) are left-aligned or centered, rather than right-aligned. Similar issues appear in Section 3.5 (Model Evaluation Index) for equations (4)-(12).
3: Figures are poorly formatted:
- Low-resolution (e.g., Figure 5’s blurred text).
- Axis labels in Figure 10 are unreadable.
- Table 3’s "√/×" notation is unintuitive; use "Yes/No" or "+/-".
4: Ensure all references are formatted consistently and properly. Some entries, such as "Passos, W. L. et al. (2023)" are not fully standardized according to the sensors reference style.
5: The article should include additional formulas to explain the SCSA attention mechanism more clearly. Furthermore, it would be beneficial to provide a more detailed rationale for selecting SCSA over other attention mechanisms. This will help readers better understand the unique advantages of SCSA in the context of the study.
6: The manuscript contains several comparative claims that lack explicit baseline references, which diminishes their scientific rigor. For instance, the statement "model parameters are reduced by 59%" (Section 4.6) fails to specify the comparison model, leaving readers to infer the reference baseline. Similar ambiguity exists in performance improvement claims (e.g., "precision increased by 7.11%" in Abstract) where the comparative benchmark is unclear.
7: The conclusions drawn from the comparative experiments in Section 4.5 lack clarity and depth. While the performance metrics (Precision, mAP50, Params, FPS, etc.) are presented in Table 4, the discussion does not sufficiently:
- Explain why MF-YOLOv10 outperforms or underperforms specific models (why does YOLOv8 achieve higher precision than MF-YOLOv10?).
- Analyze trade-offs (between accuracy and speed).
Reviewer 2 Report
Comments and Suggestions for Authors
This paper proposes using MPDIOU (a distance measure + IOU) to the SCSA-spatial channel self attention mechanism to enhance YOLOv10. The motivation for placing the SCSA attention mechanism on top of the SPPF layer of the backbone network
should be expalined
Reviewer 3 Report
Comments and Suggestions for Authors
While the introduction provides context, it might be strengthened by an even clearer articulation of the specific gap in the existing literature that MF-YOLOv10 aims to address beyond general improvements in accuracy.
While the introduction cites relevant works, ensure that the connection between these cited works and the specific innovations of MF-YOLOv10 (MPDIoU and SCSA) is clearly established. For instance, explicitly mention the limitations of existing YOLO versions or other object detection methods that MPDIoU and SCSA are designed to overcome.
To improve the reproducibility of the experiments, the Methods section could be restructured to provide more granular details. Consider separating the dataset construction, annotation, data augmentation, and training procedures into distinct subsections with more specific information
Data Augmentation: Please provide more details on the specific data augmentation techniques applied, including the parameters and ranges of transformations (e.g., rotation angle, scaling factors, brightness/contrast adjustments).
Optimizer Details: Explicitly state the optimizer used (e.g., Adam, SGD) and all relevant hyperparameters, such as momentum, weight decay, and learning rate scheduling (if any), beyond the initial learning rate. There is also a discrepancy in the learning rate mentioned in the text (0.001) and Table 1 (0.01). This needs clarification.
SCSA Module Configuration: While the functionality of the SCSA module is described, providing more specific architectural details or parameters of this module would be beneficial for readers aiming to replicate the work.
Pre-trained Model: Clearly state the specific YOLOv10n variant used as the pre-trained model and the dataset it was trained on (e.g., COCO)
Training Process: Mention the batch size used during training. Results and Analysis: Ensure that the presentation of the experimental results (Tables 2, 3, and 4) directly addresses the aims stated in the introduction. Clearly articulate how the data supports the claims of improved performance due to the proposed MF-YOLOv10 architecture. The comparative experiments (Section 4.5) effectively demonstrate the superiority of MF-YOLOv10. However, the discussion around FPS could be clearer. While YOLOv10 has a higher FPS, its accuracy is lower. Emphasize the trade-off and how MF-YOLOv10 achieves a better balance between accuracy and speed for the specific application.
Reviewer 4 Report
Comments and Suggestions for Authors
The paper is devoted to the problem of identifying piece goods in automated loading and unloading machines. The authors propose a new version of YOLOv10. Improvements are based on using new loss function and SCSA attention module. Experimental results demonstrate the efficiency of the new model.
The paper is well structured. The literature review is quite complete.
Some comments:
Line 68 – typo “Li Ke jia et al. [15]”
Please, mention the main findings of the proposed method in the Introduction.
Please, inform the reader about structure of the article at the end of Introduction section.
Please, format the article (reference list, headers, figure captions etc) according to journal requirements
Line 155 – “…dataset training… “ – obviously, model training was meant
It is not specified which data augmentation method was chosen.
Please, provide details about model convergence.
Line 216 – “BBR loss functions”, “GFLOPs.” – All abbreviations should be expanded
Equations 1-12 – Align formulas numbering to the right side
Classic IoU loss formula must be mentioned
Meaning of map50:95 is not explained
Line 313 – ti, tj , pi – subscript is missing
The captions in Fig. 8 and 9 are not informative
The quality of Fig.10 is low
Table 4 – It is not clear what is Params
Line 398 – “Compared with YOLOv5, YOLOv6, YOLOv7, YOLOv9 and YOLOv10 were 10.11%, 9.29%, 8.5%, 10.75% and 7.11% higher than YOLOv5, YOLOv6, YOLOv7, YOLOv9 and YOLOv10, respectively” – the sentence is not clear
Figure 11 is not informative at all
Round 2
Reviewer 3 Report
Comments and Suggestions for Authors
Thank you for your detailed point-by-point responses and for the careful revisions made to the manuscript.
Reviewer 4 Report
Comments and Suggestions for Authors
The paper can be published